# The Use of a Video-Assisted Intubating Stylet Technique in a Critically Ill and Contagious COVID-19 Patient

**DOI:** 10.3390/healthcare10020388

**Published:** 2022-02-18

**Authors:** Chih-Pin Yang, Hsiang-Ning Luk, Jason Zhensheng Qu, Alan Shikani

**Affiliations:** 1Department of Anesthesia, Hualien Tzuchi Medical Center, Hualien 97002, Taiwan; dingisfun903@gmail.com; 2Department of Anesthesia, Critical Care and Pain Medicine, Massachusetts General Hospital, Harvard Medical School, Boston, MA 02114, USA; jqu@mgh.harvard.edu; 3Division of Otolaryngology-Head and Neck Surgery, MedStar Union Memorial Hospital, Baltimore, MD 21218, USA; ashikani@gmail.com; 4Division of Otolaryngology-Head and Neck Surgery, LifeBridge Sinai Hospital, Baltimore, MD 21215, USA

**Keywords:** COVID-19, tracheal intubation, airway management, intubating stylet, laryngoscopy

## Abstract

Tracheal intubation and ventilatory support are among the important treatments in patients infected with COVID-19 with acute respiratory distress syndrome (ARDS) and severe hypoxia. The intubating team often uses video-assisted intubation equipment to ensure a safe and successful tracheal intubation. In this case report, we demonstrate for the first time, the use of the Shikani video-assisted intubating stylet and the Shikani intubating technique as a safe, speedy, and effective way to intubate a critically ill and highly contagious COVID-19 patient. In addition to the conventional consensus guidelines that are currently available for good practice (such as the proper use of personal protection equipment, etc.), we demonstrated that using the Shikani video-assisted intubating stylet is a reliable and proficient technique that is easy to learn while minimizing the risk of COVID-19 exposure of the airway personnel.

## 1. Introduction

Up to the 17 December 2021, there have been 271,963,258 confirmed cases of COVID-19, including 5,331,019 deaths, reported to the WHO. It is difficult to estimate how many healthcare workers (HCW) contracted COVID-19 at work, as accurate data on the characteristics and outcomes of HCW hospitalized with COVID-19 are limited. Some data indicated that 6% of adults hospitalized with COVID-19 were HCW. Among these hospitalized HCW, 28% of these patients were admitted to an intensive care unit, 16% required invasive mechanical ventilation, and 4% died [1]. A 4.5% positivity was noted out of 131,606 United States HCW who were tested at 1225 Veterans Health Administration medical centers between 12 February and 16 July 2020; however, this total may be an underestimation [2]. When personal protective equipment (PPE) is adequate, available, and properly used, infection rates of COVID-19 among HCW drop below those reported rates among the general public [3]. Unfortunately, aerosol-generating medical procedures (AGMP; e.g., tracheal intubation) are a high risk for HCW, particularly for those who manage the airway directly. In addition to the use of PPE, safe methods and protocols to conduct tracheal intubation procedures on COVID-19 patients are of paramount importance. Besides the conventional direct laryngoscopy and videolaryngoscopy, a “seeing” intubating stylet technique for tracheal intubation has been pioneered by Shikani [4] and modified by Levitan [5] decades ago, and numerous articles have been published on thousands of patients detailing the safety and advantages of the Shikani video-assisted optical stylet tracheal intubation method [6,7,8,9,10]. To our knowledge, this report describes for the first time the use of the Shikani video-assisted intubating stylet and technique to safely conduct tracheal intubation in a patient with a fatal COVID-19 infection in a negative pressure isolation room environment.

## 2. Case Presentation

A 68-year-old man with a medical history of hypertension, diabetes mellitus, and cigarette smoking, who is retired and lives an independent life, presented to our emergency room with common cold-like symptoms (weakness, running nose, and productive cough) over a period of three days. COVID-19 testing showed a positive polymerase chain reaction (PCR) with a cycle threshold (Ct) of 26. He was immediately admitted to the isolation ward.

TOCC reporting (travel, occupation, contacts, clusters) alarmingly revealed that this patient had had positive contact history with confirmed COVID-19 cases and involved clusters with his family members who were later also diagnosed with COVID-19 infections. During the quarantine period, fever (38.4 °C) was noted. Vital signs were stable: blood pressure 135/64 mmHg, heart rate of 90 beats/minute, respiratory rate of 20/minute, and oxygen saturation (SpO_2_) 94% on room air. Pulmonary auscultation revealed crackles over both lungs, but the general physical examination was otherwise unremarkable. The chest X-ray showed diffuse increased interstitial marking, patchy consolidations, and ground-glass opacities of both lungs (serial chest radiography, see Figure 1A–D). Ertapenem was prescribed to cover the pneumonia, but unfortunately his symptoms worsened on Day 2. Additional treatment modalities against COVID-19 were implemented, including dexamethasone, remdesivir, and tocilizumab. High-flow nasal cannula (HFNC) oxygen therapy was started when SpO_2_ dropped down 89%. On Day 4, as his hypoxia worsened, tracheal intubation was deemed necessary for ventilator support.

Because of the active COVID-19 infection and potential for contagion, a special task force organized by anesthesiologists (one attending physician and one resident) was assigned to perform the tracheal intubation. Prior to the mission, an infection-control training and airway management rehearsal based on the available consensus guidelines were conducted. Both airway managers wore adequate PPE (Figure 1E). The intubation task was executed in a negative pressure isolation room (Figure 1G). The plan was to intubate the patient following the Shikani intubating technique using a video-assisted intubating stylet (manufactured by Trachway^®^, Markstein Sichtec Medical Corp, Taichung, Taiwan). A modified rapid sequence intubation protocol was adopted, including pre-oxygenation with HFNC, atropine, ketamine, midazolam, etomidate, rocuronium, and succinylcholine (Table 1). After the patient was sedated and 60 s after the neuromuscular blocking agents became effective, a cuffed oral endotracheal tube (diameter 7.5) was swiftly and safely inserted into the larynx and trachea within 7 s (counting time “from lip to trachea”; Figure 1F and Figure 2; also see the Appendix A). The intubating process was documented using a video screen module (video clip, please see Appendix A). The whole process was smooth, without any significant hypoxemic episode. The respiratory therapists then took over for the ventilator settings (pressure control mode, FiO_2_ 60% with pressure support 19 cmH_2_O, and positive end-expiratory pressure 7). The airway managers left the isolation room and took off the PPE according to the standard doffing procedure.

After an initial period of stabilization, the patient’s pneumonia and respiratory condition kept on deteriorating and a tracheotomy had to be performed on Day 23 of admission. The patient developed full acute respiratory distress syndrome, continued a progressive downhill course and tragically passed away on Day 28. It is worth mentioning that although dexamethasone, remdesivir, and tocilizumab had been given to this patient, he unfortunately developed secondary pulmonary infections. Positive sputum cultures (for *Klebsiella pneumoniae*, *Pseudomonas aeruginosa*, and *Stenotrophomonas maltophili*), positive blood cultures (*Staphylococcus epidermidis*), positive serum *Aspergillus* antigens, and herpes simplex virus DNA were noted. Respiratory failure and septic shock lead to multi-organ failure and eventual death.

## 3. Discussion

This is the first report discussing the use of the Shikani video-assisted optical stylet and Shikani tracheal intubation technique to safely secure the airway in a critically ill and highly contagious COVID-19 patient who needed mechanical ventilator support.

According to data collected by the World Health Organization (WHO) prior to the availability of COVID-19 vaccines, 14 percent of COVID-19 cases reported to the WHO were among HCW [11]. Several studies have evaluated the seroprevalence of infection in HCW, which appears to differ by region. In one study from Germany, the seroprevalence in 316 HCW was found to be 1.6 percent [12]. By contrast, in Belgium, the seroprevalence was 6.4 percent in a sample of 3056 HCW [13]. Although it is not possible to know exactly how and where those HCW contracted COVID-19, it is clear that the highest risk is among HCW whose responsibility is to secure the airway of infected patients. This includes procedures such as tracheal intubation, tracheostomy, presumed for emergency front-of-neck airway, non-invasive ventilation, mask ventilation, disconnection of ventilatory circuits during use, tracheal extubation, cardiopulmonary resuscitation, bronchoscopy, and tracheal suction without a ‘closed in-line system.’

Immediately after the outbreak of COVID-19, several clinical principles and guidelines were soon developed to guide intubators (airway operators or managers) for tracheal intubation in patients with suspected and confirmed COVID-19 infections [14,15,16,17]. Agreed guidelines require wearing full PPE at all times, a long-sleeved gown, double gloving, goggles, eyewear face shield, boots, and respirators (e.g., N95 mask, powered air-purifying respirator) ideally in a negative pressure environment, if available. Figure 1E shows our airway management team dressed up before entering the negative pressure room and conducted the tracheal intubation task inside the room (Figure 1F,G). Sequence for donning and doffing PPE was in compliance with the guidelines.

The adequacy and availability of PPE supply varied tremendously among countries and facilities during the course of the pandemic; however, the situation changed over time with the improvement in supplies and logistics. Because the virus is highly concentrated in the tracheal airway, intubation continues to be a high risk and challenging procedure. Therefore, we feel that consensus on the guidelines for COVID-19 airway management should be established in order to enhance the safety of the HCW [14,15,16,17], and our argument is that the video-assisted intubating stylet technique should be part of the intubating guidelines [18,19,20]. In general, we recommend that experienced intubators, who are comfortable with various airway management techniques, handle the airway of ill COVID-19 patients. Shortening the intubation time will undoubtedly minimize the risk to the HCW, the patient, and inexperienced operators [18,19,20].

To date, in the world of the COVID-19 pandemic, conventional laryngoscopy intubation techniques have not been satisfactory [16]. Successful intubation at the first attempt was reportedly around 85% and lower. The time to succeed in intubating was on the scale of 3 min and 5 min. The multiple attempts of intubation and prolonged duration to complete intubation could partially contribute to a high incidence of post-intubated hypoxemia events (10 to 30%).

Video-assisted intubating stylet technique, using the Shikani method of intubation [4], has been gaining popularity in many medical centers in Taiwan and around the world in recent years [6,7,8,9,10]. Studies in the literature have shown that the mean time to carry out intubation when using the Shikani stylet was 42.4 s [21] and 58 s [22]; however, this includes operators who are still in the learning curve and/or are not quite experienced with the Shikani stylet. More than 90% of tracheal intubations in daily practice, e.g., in our hospital, are achieved by the Shikani intubating stylet technique [23,24,25]. Because of our expertise and experience with the device, the time to carry out intubation in our hands ranged from 4 to 10 s, and our comfort level with the device makes us confident enough to consider shifting to the Shikani intubating stylet and technique in a negative pressure room for intubations of all COVID-19 patients. As shown in Figure 2 and in the Appendix A, we smoothly and swiftly accomplished tracheal intubation (within 7 s) and secured the airway uneventfully. It is worth mentioning that there have been only 16,816 confirmed cases since the outbreak of COVID-19 in Taiwan. However, between May and June 2021, there was an unexpected surge of COVID-19 cases in Taipei metropolitan areas and caused more than 800 deaths during that time. It is estimated that there were around 400 patients in Taiwan with COVID-19 who needed to be intubated and received ventilatory support during this surge crisis. Most tracheal intubations were conducted using videolaryngoscopy, few were carried out with conventional direct laryngoscopy. In our hospital, there were only six patients with severe COVID-19 infections who needed tracheal intubation, and this was carried out in a negative pressure isolation room. In the last three patients, we shifted to the Shikani video-assisted intubating stylet and technique as we felt it to be faster and safer, and at present it is our intubation method of choice. The entire process of tracheal intubation using the intubating stylet is described in Figure 1F and Figure 2 (also see the Appendix A). Finally, it is worth stressing that, although certain improvised barrier enclosure devices seem promising [23,24,25,26], it remains to be evaluated in the real-world scenario. Before that, adequate PPEs and a reliable intubating tool/technique should be cautiously adopted.

In conclusion, the authors find, from their experience, that the video-assisted intubating stylet technique is a reliable and effective airway management modality for patients who are critically ill and have contagious COVID-19 infections. In addition to being familiar with the proper use of PPEs, we recommend that HCW who are responsible for the airway management of patients with COVID-19 have enough training to become comfortable with the Shikani seeing stylet and proficient in the Shikani technique as this will shorten the time of intubation and minimize the risk of exposure.

## Figures and Tables

**Figure 1 healthcare-10-00388-f001:**
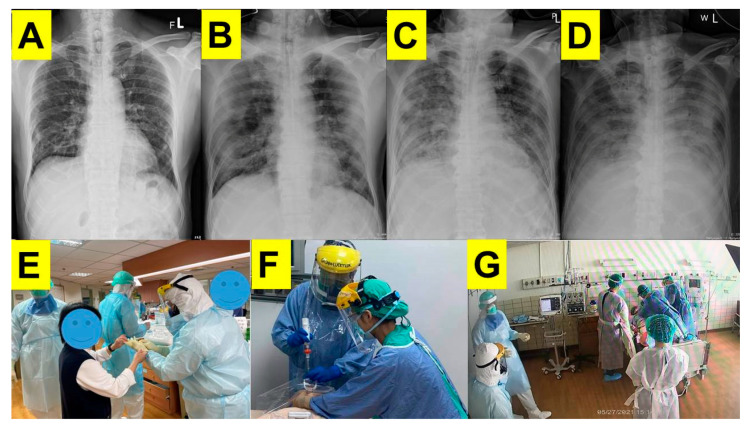
Serial chest radiography taken on Day 1 (**A**); Day 6 (**B**); Day 19 (**C**); Day 26 (**D**). Donning of PPE before entering the negative pressure isolation room (**E**). An illustrated image was reconstructed to demonstrate how tracheal intubation using a video-assisted intubating stylet technique had been performed in the isolation room (**F**). After tracheal intubation was completed smoothly, the respiratory therapy team took over for further medical care (**G**).

**Figure 2 healthcare-10-00388-f002:**
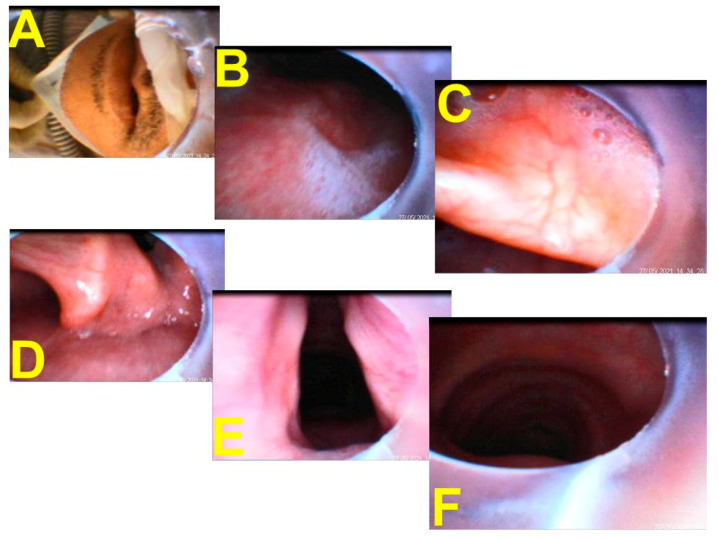
A serial pictorial illustration of tracheal intubation using video-assisted intubating stylet technique in a patient with a severe COVID-19 infection. (**A**) Ready to insert the intubating stylet into the patient’s oral cavity while he still received HFNC treatment. (**B**) Some mucus and saliva can be seen in the pharynx. Close views on vallecula (**C**) and corniculate and cuneiform cartilages (**D**). A clear and full view of the glottis before inserting the endotracheal tube (**E**). (**F**) Tracheal rings are clearly seen and then the tip of the endotracheal tube is inserted into the trachea before pulling out the intubating stylet.

**Table 1 healthcare-10-00388-t001:** Time sequence of the tracheal intubation procedure.

Events	Time	Conditions
The on-duty airway management team (one senior anesthesiologist and one resident doctor) was called up and briefed.	20 min	Institutional and departmental preparations have been in place during the COVID-19 pandemic.
Donning of PPE	15 min	Wear full PPE, assisted by a buddy system in the nurse station of the ICU
Pre-oxygenation	Continuously	This patient already received HFNC oxygenation in the negative pressure isolation room.
Bolus injection with atropine, ketamine, midazolam, and etomidate	60 s	HFNC was still on
Bolus injection of rocuronium and succinylcholine	60 s	No face mask ventilation was used. Apneic insufflation with HFNC oxygenation was continuously applied.
Tracheal intubation	7 s (from lip to trachea). Please see the video clip in the Appendix A.	A video-assisted intubating stylet technique was used. The airway assistant facilitated mouth opening and jaw thrust. No cricoid maneuver.
Airway confirmed and ventilator set up.	30 s	Assisted by ICU nurses and respiratory therapist.
Doffing of PPE	10 min	Observed by an infection control staff

COVID-19: Coronavirus disease 2019. PPE: Personal protective equipment. HFNC: High-flow nasal cannula. ICU: Intensive care unit.

## Data Availability

Not applicable.

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
