# Peer review of "The Use of a Video-Assisted Intubating Stylet Technique in a Critically Ill and Contagious COVID-19 Patient"

_healthcare, 2022, doi:10.3390/healthcare10020388_

Round 1

Reviewer 1 Report

Comments and Suggestions for Authors
This is a case report.
The group has reviewed several research articles about the different type during intubation procedure. The authors also briefly mentioned about the recent developments and possible future plans/improvements in the research field.

1. It is a very relevant topic, considering the COVID-19 pandemic.
2. It is a well written manuscript, and the sections are well organized and include the relevant topics nevertheless some recommendations should be considered by the authors.
3. I have found a very interesting, recent (2021), very well cited and informative paper. Please check a very recent paper” Combating COVID-19 during Airway Management: Validation of a Protection Tent for Containing Aerosols and Droplets” Appl. Sci. 2021, 11(16), 7245; https://doi.org/10.3390/app11167245.
4. Please add the table for step by step about the intubation procedure and time/condition.

Author Response

Comments and Suggestions for Authors (Reviewer-1)

This is a case report. The group has reviewed several research articles about the different type during intubation procedure. The authors also briefly mentioned about the recent developments and possible future plans/improvements in the research field.

Comment-1:  It is a very relevant topic, considering the COVID-19 pandemic.

Response-1: Thank you.

Comment-2:  It is a well written manuscript, and the sections are well organized and include the relevant topics nevertheless some recommendations should be considered by the authors.

Response-2: Thank you. We have reorganized and expanded the contents of the manuscript in order to explain our views in a better way.

Comment-3: I have found a very interesting, recent (2021), very well cited and informative paper. Please check a very recent paper” Combating COVID-19 during Airway Management: Validation of a Protection Tent for Containing Aerosols and Droplets” Appl. Sci. 2021, 11(16), 7245; https://doi.org/10.3390/app11167245.

Response-3: Thanks for your referring to this interesting article. Indeed, among various barrier designs, the plastic sheet tent might play some roles in protecting the intubators during the aerosol-generating medical procedures. We have added this paper as the reference-25 and put the context in the manuscript.

“Finally, it is worthy to stress that, although certain improvised barrier enclosure devices seem promising [22-25], it remains to be evaluated in the real world scenario. Before that, adequate PPEs and a reliable intubating tool/technique should be cautiously adopted.”

  1. Weng, C.H.; Chiu, P.W.; Kao, C.L.; Lin, Y.Y.; Lin, C.H. Combating COVID-19 during airway management: Validation of a protection tent for containing aerosols and droplets. Appl. Sci. 2021, 11, 7245. https://doi.org/10.3390/app11167245

Comment-4: Please add the table for step by step about the intubation procedure and time/condition.

Response-4: A new Table was added to list the intubation procedure with references of time and condition of the process.

Events

Time

Conditions

The on-duty airway management team (one senior anesthesiologist and one resident doctor) was called up and briefed.

20 min

Institutional and departmental preparations have been in place during COVID pandemic.

Donning of personal protective equipment (PPE)

15 min

Wear full PPE, assisted by a buddy system in the nurse station of the intensive care unit.

Pre-oxygenation

Continuously

This patient already received high flow nasal cannula (HFNC) oxygenation in the negative pressure isolation room.

Bolus Injection with atropine, ketamine, midazolam, and etomidate

60 sec

HFNC was still on

Rocuronium and succinylcholine

60 sec

No face mask ventilation was used. Apneic insufflation with HFNC oxygenation was continuously applied.

Tracheal intubation

7 sec (from lip to trachea). Please see the video clip in the supplementary material.

Using a video-assisted intubating stylet technique. The assistant facilitated mouth opening and jaw thrust. No cricoid maneuver.

Airway confirmed ventilator set up.

30 sec

Assisted by ICU nurse and respiratory therapist.

Doffing of PPE

10 min

Observed by an infection control staff

Reviewer 2 Report

The article does not add anything new to the literature on the subject. The use of video-laryngoscopes is even more common in infectious patients. 

Author Response

Comments and Suggestions for Authors (Reviewer-2)

Comment-1: The article does not add anything new to the literature on the subject. The use of video-laryngoscopes is even more common in infectious patients.

Response-1: Thanks for your viewpoint. In this article, we wish to add something new, if there is any, about the role of video-assisted intubating stylet technique on severe COVID-19 patient during the pandemic. To our knowledge, this is the first clinical report regarding the use of such unique airway technique in such a highly risky clinical situation.

During the last decades, the role of videolaryngoscopy is indeed kind of paradigm shift for clinical practices and contributes significantly to the airway managers under various clinical scenarios. This notion is supported by the published airway management guidelines (references 14-16 in the manuscript) and put the vidolaryngoscopy into the priority role during tracheal intuabtion on COVID patients. Meanwhile, with tremendous clinical experience, skills, and overwhelming preference of using video-assisted intubating stylet technique (also called Shikani technique) in Taiwan for more than a decade, we confidently and comfortably handle airway management in various clinical scenarios, including the tracheal intubation on COVID patients in the negative pressure isolation room environment (references 22-24 in this manuscript).

We acknowledge this clinical report is not a comparative clinical study among the role of direct laryngoscopy, videolarygoscopy and video-assisted intubating stylet technique in COVID situation (which is almost impossible to conduct such kind of clincal trial under COVID scenario, no matter designed as prospective or retrospective in nature). We did not intend to ignor the“more common”reality of using direct laryngoscopy and videolaryngoscopy, either during normal lifetime or COVID pandemic. Instead, we would like to present, for the first time, the advantages and benefits of using video-assisted intubating stylet technique on COVID patients, based on the principles of being swift, accurate, and safe.

Reviewer 3 Report

In this study, the authors report the first use of the video-assisted intubation stylet technique in a patient with fatal COVID-19.   Although the authors report an interesting technique for airway management, I think the article needs to be improved to better discuss the pros and cons of this technique in patients with COVID-19.  The authors should focus on the following questions:  - Why do they specifically recommend this technique over videolaryngoscopy? Please explain using references.  - Why more so in COVID-19 patients than in patients with respiratory distress.    Minor comments:    1. Do you know the viral load of SARS-CoV-2 during the intubation procedure? I think the patient's cycle threshold (Ct) value would be appropriate information. 2. Do you have SPo2 data the minutes after intubation?  Could you provide the time to intubate the patient?    2. Please indicate if the patient had foreseeable intubation difficulties.    In the summary, I suggest that you revise the last sentence.  In this case report, the design of the study does not allow you to demonstrate anything, you are only reporting your experience and view of a technique that appears to be safe and effective.    Line 135, could you explain why other techniques are not satisfactory? 

Author Response

Comments and Suggestions for Authors (Reviewer-3)

In this study, the authors report the first use of the video-assisted intubation stylet technique in a patient with fatal COVID-19.

Although the authors report an interesting technique for airway management, I think the article needs to be improved to better discuss the pros and cons of this technique in patients with COVID-19. The authors should focus on the following questions:

Comment-1: Why do they specifically recommend this technique over videolaryngoscopy? Please explain using references.

Response-1: Thank you very much for your constructive comments and valuable suggestions on our manuscript. We have revised the Discussion section of the manuscript accordingly (especially the last three paragraphs in this section).

As the reviewer indicated, this case report was not designed as a comparative clinical study. We proposed the advantages of using Shikani technique in this particular occasion are the excellent first-attempt success rate (100%) and very prompt time to intubate (7 sec). A numerous clinical studies have been done to demonstrate the advantages of Shikani technique in various clinical scenarios (here we only cited references 20 and 21). Meanwhile, we cited the reference-19 to infer that conventional laryngoscopy (and perhaps videolaryngoscopy) might not be advantageous under the clinical scenarios like COVID, based on the findings of time-to-intubate and first-attempt success rate. 

Comment-2: Why more so in COVID-19 patients than in patients with respiratory distress.

Response-2: We demonstrated that the use of video-assisted intubating stylet technique (Shikani technique) be a valid, efficient, and comfortable tracheal intubation technique in both COVID-19 patients and others with respiratory distress. The intubation technique is proficient and has come a favorable video-assisted option in our medical center and many others in Taiwan in the last decade. This intubation technique is suitable for both elective airway management and some difficult intubation scenarios (e.g., obstructive upper airway, morbid obesity, confined cervical spine motility, etc).

Comment-3: Do you know the viral load of SARS-CoV-2 during the intubation procedure? I think the patient's cycle threshold (Ct) value would be appropriate information.

Response-3: Excellent point! This patient was an index patient for a family cluster infection in Hualien county during the outbreak of COVID-19 in May, 2021. It was found that the patient’s Ct value was 26. We added this information in the revised manuscript.  

Comment-4: Do you have SPo2 data the minutes after intubation? 

Response-4: The SpO2 was around 94% before tracheal intubation (with HFNC oxygenation). During the apneic period (60 sec, waiting for achieving adequate neuromuscular blocking action by NMBAs), the patient’s saturation dropped down to 89% and immediately went back to 94% after the tracheal intubation. It should be emphasized that we continued the HFNC oxygenation until the moment of intubation. We planned to use mask ventilation during induction only when unbearable arterial desaturation occurred.  

Comment-5: Could you provide the time to intubate the patient?

Response-5: The time to intubation was 7 sec in this patient (displayed from the video recordings shown in the supplementary material). We added this information in the revised text.

Comment-6: Please indicate if the patient had foreseeable intubation difficulties. 

Response-6: Although MACOCHA score is not widely use but recommended by the consensus guidelines for airway management on COVID patients. In this patient, the only apparent risk factor was “hypoxemia”. Therefore, a foreseeable difficult intubation was not indicated in this patient.

Comment-7: In the summary, I suggest that you revise the last sentence. In this case report, the design of the study does not allow you to demonstrate anything, you are only reporting your experience and view of a technique that appears to be safe and effective.

Response-7: Thank you for your opinion and advice. Very fair and helpful. We have revised our statement in the Summary accordingly.

Comment-8: Line 135, could you explain why other techniques are not satisfactory?

Response-8: Thanks for your question, regarding the statement of “To date, in the world of COVID-19 pandemic, conventional laryngoscopy intubation techniques have not been satisfactory [16]. Successful intubation at first attempt was reportedly around 85% and lower. Time to succeed in intubating was in the scale of 3 min and 5 min. The multiple attempts of intubation and prolonged duration to complete intubation could partially contribute to high incidence of post-intubated hypoxemia events (10 to 30%).”

For COVID-19 patients who need tracheal intubation, one of the difficulties encountered during the intubating process is desaturation. As the real-world data shown in the reference-16, when videolaryngoscopy and direct laryngoscopy were used as the intubating tools, the “first-attempt success rate” was lower than 85% and “time to intubate” could be as long as 3 to 5 min. This unsatisfactory result could be due to many factors (e.g., high mental load of the intubators, obscured vision and decreased dexterity by the PPEs, existing difficult airway, quick desaturation, etc). The goal of tracheal intubation for COVID patients should be (as coined by the consensus guidelines) swift, accurate, and safe. We reported this intubating modality (e.g., Shikani technique) as an alternative for video-assisted intubation devices.

Round 2

Reviewer 2 Report

the authors' answers are not satisfactory to me. It is a case study - and the legitimacy of using videolaryngoscopy has already been presented in many articles. This article does not bring anything new to science.

Author Response

Thanks for your sincere and straight comments and response on our manuscript. In addition to the role of videolaryngoscopy, it is our intention to bring up with another thought on clinical applicaiton of video-assisted intubating stylet technique on various clinical scenarios, including this challenging one (COVID-19 scenario). We really appreciate your expertise view on this matter. 

Reviewer 3 Report

In this review of the article, the authors address most of my concerns. 
Just pay attention to the font used in Table 1, which is different from the rest of the article.

Author Response

Thanks again for your professional opinions and constructive comments on our manuscript. We will definitely take care of the fonts of the Table-1 during proofing process.